# Effects of Breed, Exercise, and a Two-Month Training Period on NT-proBNP-Levels in Athletic Dogs

**DOI:** 10.3390/ani13010016

**Published:** 2022-12-20

**Authors:** Frane Ivasovic, J. Novo Matos, Michele Wyler, Tony M. Glaus

**Affiliations:** 1Division of Cardiology, Clinic for Small Animal Internal Medicine, Vetsuisse Faculty, University of Zürich, 8006 Zurich, Switzerland; 2Puregene AG, 4314 Zeiningen, Switzerland

**Keywords:** biomarker, natriuretic peptide, athletic dog, cardiology

## Abstract

**Simple Summary:**

In people, different types of physical activity can induce various types of heart adaptation. These changes are not only associated with morphological and physiological consequences on the heart, but they are also accompanied by alterations in cardiac biomarkers. In dogs, these parameters are clinically used, but they have not been thoroughly evaluated in animals that are regularly engaged in high-level sporting activities. The aim of this paper was to describe the effects on a specific cardiac blood parameter due to the physical activity in two different breeds of dogs in their respective sports. Additionally, our study included the use of echocardiography and electrocardiography to further assess these parameters. Exercise induced different responses in the two groups, and the measured parameter correlated with certain echocardiographic values. Our findings suggest that the physical activity must be considered when evaluating this blood parameter in canine patients. Further studies are necessary to evaluate the utility of this biomarker to distinguish possible cardiac diseases from physiological athletic adaptations.

**Abstract:**

N-terminal pro-b-type natriuretic peptide (NTproBNP) is a cardiac biomarker used to detect myocardial wall stress. Physical activity and cardiac disease can affect serum NTproBNP concentrations. In people, different types of physical activity have different effects on NTproBNP. Our hypothesis was that physical activity and training have an effect on NTproBNP concentrations depending on the type of exercise and the intensity. Seven German Shepherd dogs (GSD) under military training performing short bursts of fast-paced interval exercise and seven Eurohounds (EHs) training for racing competition with endurance exercise were included in the study. Blood samples were taken at enrollment (T_0_) and after a two-month (T_2mth_) training period; on both occasions, the samples were acquired before and after physical exercise. An echocardiographic evaluation was performed at T_0_. Echocardiographic heart size was larger in the EHs compared to the GSDs. The NTproBNP concentration was higher in the EHs than in the GSDs before and after exercise at T_0_ and T_2mth_. Echocardiographic parameters of heart size and wall thickness correlated with NTproBNP at T_0_ before and after exercise. Exercise induced an elevation of NTproBNP in the EHs at T_0_ and T_2mth_, while in the GSDs this was observed only at T_0_. In the EHs, post exercise was associated with higher NTproBNP at T_2mth_ compared to T_0_, while in the GSDs the opposite pattern was noticed. From our study, the serum NTproBNP concentration differs between breeds. Intense physical activity causes an increase in NTproBNP. A two-month training period does not affect the NTproBNP concentration at rest. Intense physical activity may increase NTproBNP above the reference range in individual dogs.

## 1. Introduction

N-terminal pro-b-type natriuretic peptide (NTproBNP) is a cardiac biomarker used as a diagnostic tool for identifying and staging cardiac disease in dogs [1,2,3,4,5]. It is naturally found in atrial and ventricular myocardium. Increased myocardial stretch, specifically myocardial wall stress, causes increased NTproBNP production and consequently increased serum levels [6]. Besides primary cardiac disease, various extracardiac stimuli can result in an increased serum concentration of natriuretic peptides, e.g., hypoxia and alpha- and beta-adrenergic agonists [7,8]. A high serum concentration can also result from decreased excretion, for instance in renal disease [9,10]. Furthermore, serum concentration is affected by breed, age, and sex [11,12,13]. Finally, serum concentrations fluctuate over time [14]. Despite a high-reported sensitivity and specificity of elevated serum levels for identifying hemodynamically relevant cardiac diseases [2], these extracardiac factors may confound the interpretation of circulating concentrations and affect its diagnostic utility.

Physical activity has also been shown to result in changes of cardiac biomarker concentrations. One reason to measure natriuretic peptides associated with physical exercise and the effect of training is in an attempt to differentiate physiologic cardiac changes associated with athletic activity from pathological cardiac changes. There are multiple studies in human medicine with the focus on the type of exercise, level of training, and association with physiologic versus pathologic cardiac hypertrophy [15,16,17,18]. In brief, NTproBNP was not different in healthy endurance athletes with athlete’s hearts and untrained healthy control subjects [15]. Strenuous as well as only moderate exercise was associated with an increase in NTproBNP, but not usually beyond the reference range [16]. Furthermore, whereas NTproBNP was similar in healthy runners and healthy control subjects at rest and after exercise with only a marginal increase after exercise, it showed a 10-fold increase after exercise in patients with hypertension and secondary ventricular hypertrophy. Finally, the same study showed that in athletes with echocardiographic primary hypertrophic cardiomyopathy, NTproBNP was already 10-fold increased at rest and increased further after exercise [16]. Post-exercise NTproBNP levels did not correlate with serum lactate concentrations [17]. These studies suggested that NTproBNP measured before and after exercise might be useful to screen for pathologic forms of left ventricular hypertrophy. In another study, the NTproBNP was different before but not after strenuous exercise in athletes with gray-zone LVH compared to athletes with HCM [18]. Thus, results of different studies are not consistent.

In dogs, several studies have been performed evaluating the effect of physical exercise on various laboratory biomarkers. In Greyhounds performing a 7 km high-intensity run, cTnI concentrations increased significantly [19]. In sledge dogs after high-intensity exercise, serum cTnI and C-reactive protein were mildly increased [20]. The effect of exercise on inflammatory biomarkers was further studied in Basset hounds, and a significant increase in prostaglandin E_2_ values was found [21]. There are also studies on the effect of physical activity on plasma NTproBNP concentration [22,23]. Strenuous exercise in working farm dogs caused an increase in NTproBNP, but it was less pronounced on each consecutive day over a four-day period [22]. Finally, NTproBNP was higher in dogs with preclinical degenerative mitral valve disease at rest and showed a more pronounced increase after exercise compared to healthy control dogs, but values remained within the reference range [23].

The authors have identified in their clinical practice racing dogs with highly successful profiles in their competitive careers, which had echocardiographic changes suggestive of LV volume overload and decreased systolic function, judged by the authors as an athletic heart phenotype. Moreover, some of these dogs had markedly high NTproBNP levels. One of these dogs was a European Hound (EH), a mixed breed between Alaskan husky and German shorthaired pointer, purpose-bred for sledge racing competition, i.e., endurance sports. These observations have motivated the present study.

The hypotheses of the study were (1) that the EHs have a different echocardiographic phenotype and higher NTproBNP serum concentrations than a control breed (German Shepherd Dog, GSD) also used for intensive physical activities, (2) that acute myocardial stretch caused by a short interval of high-intensity exercise [24] in the GSDs would result in a more marked NTproBNP elevation than endurance exercise in the EHs, and (3) that several weeks of training would affect NTproBNP in both breeds and scenarios. As markers of exercise intensity, serum lactate was measured and the heart rate was monitored by a Holter recording device.

## 2. Materials and Methods

### 2.1. Study Population

The study protocol was ethically approved by the veterinary office, State of Zurich (ZH260/16). Two groups with seven dogs each composed the study population. The first group consisted of seven GSDs starting training as military attack dogs. Of the seven dogs, six were intact males and one a spayed female, the median age was three years (range 2–4 years), and the median body weight was 31.5 kg (29–36 kg). They were considered healthy based on a complete physical examination and routine biochemical and hematological analyses and had to pass behavioral tests. These dogs had not received special physical training up to this point.

The second group consisted of seven EHs (two intact males, four neutered males, and one intact female) used for sledge racing. The median age of these dogs was six years (range 1–10 years) and the median body weight was 28.5 kg (range 20–35 kg). With one exception, all dogs in this group were considered healthy based on physical and echocardiographic examinations. One dog (EH_α_, 32 kg) with outstanding sports results on competitive activities had repeatedly shown a markedly elevated NTproBNP (>3218 pmol/L); echocardiographically he had moderate left ventricular volume enlargement considered to reflect an athletic heart. His results are shown separately from the other EHs.

These seven EH dogs’ usual training sessions consisted of running at a high pace while pulling a weight for intermediate distances (5–15 km). Before entering the study, they underwent a three-week period without training to obtain baseline values.

### 2.2. Study Protocols

The protocol for the GSD group was the following: on day 1 (T_0_), each dog underwent a physical and an echocardiographic examination, a first blood sample was drawn, and a Holter-ECG was applied. The exercise program consisted of three parts. First, the dog was lying on the ground and had to bite-attack the trainer in a designated area on-demand. Second, the dog had different targets at disposition at a distance of 20 m; the trainer gave the order to attack one specific target and subsequently recalled the dog. Third, the dog had to run a 30 m obstacle course before catching and blocking a running obstacle. The duration of the whole exercise was around 15 min. After the last activity, a second blood sample was obtained, and the Holter recording device was removed 15 min later. These exercises were performed daily for five days a week, and dogs were reassessed with the same protocol after two months (T_2mth_).

The protocol for the EH group was the following: on day 1 (T_0_), each dog obtained a physical and an echocardiographic examination, and a first blood sample was drawn. Four dogs were attached to a sledge and had to pull 70 kg on a trailer for 5 km in 10 to 12 min; after completion of the run, the second blood sample was drawn. Dogs were trained regularly, by performing different distances of sledge trails or running by being attached directly to the runner; these dogs were reassessed as well after two months (T_2mth_). In this group, the echocardiographic evaluation was performed only at T_0_, while the Holter examinations were only performed at T_2mth_. The Holter was applied after the initial blood sampling and the recording device was removed 15 min after the completion of the run.

### 2.3. Echocardiography and Holter-ECG

All echocardiographic examinations were performed in standard right and left lateral recumbency [25] by one operator (TMG) (GE Logiq Q with an M4S Matrix Probe, GE Healthcare, Glattburg, Switzerland). All echocardiographic measurements were performed offline from the digitally stored cine-loop recordings by another operator (FIVA). The following echocardiographic parameters obtained from M-Mode measurements using the leading-edge-to-leading-edge method [26] were used for statistical analysis: left ventricular internal diameter in diastole (LVIDd), left ventricular internal diameter in systole (LVIDs), their normalized values (LVDDN and LVDSN) as previously described by using the formulas: LVDDN = LVDd/Body Weight^0.294^ and LVDSN = LVDs/Body Weight^0.315^ [27], interventricular septum thickness in diastole (IVSd), left ventricular free wall thickness in diastole (LVWd). Additionally, the ratio between the LVWd and LV internal radius (h/R), defined as h/R = LVWd/(½LVDd), was calculated [28]. Similarly, the relative wall thickness (RWT) of the left ventricle as the ratio of the sum of the IVSd and LVWd to LVIDd was calculated [29]. The left atrium to aortic root (LA/Ao) ratio was determined in the 2-dimensional right-sided short axis view in early diastole [30].

Holter ECGs were recorded with Lifecard CF and analyzed offline with commercial software (Sentinel Version 9.0.2, Spacelabs Healthcare, Hertford, UK) by one operator (FIVA). Recordings were reviewed for the presence of rhythm abnormalities and to obtain mean heart rates before, during and after exercise. The baseline heart rate (HR_Base_) was calculated from a ten-second strip at rest, two minutes after the Holter recording start, i.e., after a short period of acclimatization to the Holter attachment. The maximal heart rate during activity (HR_Max_) was defined as the maximal ten-second heart rate during physical activity. Heart rate was again obtained based on a 10 s period, 10 min after the end of the exercise (HR_Rec_).

### 2.4. Blood Sample Collection

A venous blood sample (3 mL) was collected into ethylenediaminetetraacetic acid (EDTA) tubes from each dog’s cephalic vein immediately before and after physical activity on both examination days. Blood plasma was obtained after centrifugation (5000× *g* for 15 min) immediately after collection; it was placed in (EDTA) tubes and kept on dry ice. Part of the collected blood was stored in tubes containing sodium fluoride. The frozen samples were sent to determine the NTproBNP concentrations (Cardiopet ELISA, IDEXX Diavet, Bäch, Switzerland); fluoride plasma was used for the lactate measurements (IDEXX, Diavet, Switzerland).

### 2.5. Statistical Analysis

Statistical analyses were performed by commercially available software (IBM, SPSS Statistics, Version 23; R Core Team 2018, v.3.5.1 on RStudio Inc., version 1.29 Boston, MA, USA). All data underwent an initial descriptive assessment to reveal distributions and the presence of outliers. Difference scores were generated, and under visual inspection, statistical outliers were identified. Data were analyzed for normal distribution by the Shapiro–Wilk test and visually assessed via qqplots. The t-test was used for normally distributed and Wilcoxon signed-rank test for not normally distributed data. To account for the repeated measurements of the single dog, we used the paired versions of the previously described tests when appropriated. For NTproBNP and lactate, in addition to comparing the absolute values in the different breeds and at the various time points, the difference of the values induced by the exercise was calculated (ΔNTproBNP, ΔLactate). The Pearson’s correlation was used to test the association between the different echocardiographic parameters and the NTproBNP values, as well between the variation in heart rate (ΔHR) and the variations in NTproBNP (ΔNTproBNP). For all calculations, statistical significance was set at a *p*-value < 0.05. Data are given as median (range).

Given the observation that the exceptional dog EH_α_ was not representative of the EH, but a statistical outlier in most respects, all calculations within and between the breeds were performed by excluding EH_α_’s data.

## 3. Results

### 3.1. Echocardiography and Holter Examination

The GSD group had a significantly smaller normalized ventricular diameter than the EH group (median LVDDN at T_0_, GSD 1.4, EH 1.65 *p* = 0.01); the other echocardiographic parameters were not different between the two groups (Table 1). Heart size did not change in the GSD group during the 2-month training period.

At T_0_, the quality of the Holter examinations was good in all GSDs. No pathological arrhythmias were detected. The HR_Base_ was 100 (80–160); the HR_Max_ was 240 (116–300) bpm, and the HR_Rec_ was 100 (100–150). The HR_Max_ was significantly higher than the HR_Base_ (*p* = 0.022) and the HR_Rec_ (*p* = 0.022); there was no difference between the HR_Base_ and HR_Rec_ (*p* = 0.783).

At T_2mth_, the quality of the Holter examination in 1/7 GSD dogs was poor and excluded from analysis. No pathological arrhythmias were detected. The HR_Base_ was 132 (102–156) bpm, the HR_Max_ was 249 (180–270) bpm, and the HR_Rec_ was 141 (120–168) bpm. Maximal heart rates were significantly higher than the baseline in the GSD group (*p* = 0.031). Recovery heart rates were significantly lower compared to HR_Max_ (*p* = 0.031), but not different from HR_Base_ (*p* = 0.343). There were no differences in heart rates in the GSD group between T_0_ and T_2mth_.

In the EHs, the HR_Base_ was 114 (60–180) bpm, the HR_Max_ was 225 (168–320) bpm, and the HR_Rec_ was 141 (108–144) bpm. Maximal heart rates were not significantly higher than the baseline (*p* = 0.063). Recovery heart rates were significantly lower compared to HR_Max_ (*p* = 0.031), but not different from HR_Base_ (*p* = 0.419). The median heart rates were not different between the breeds at the baseline (*p* = 0.48), at the maximal heart (*p* = 0.74) rates, and after recovery (*p* = 0.25) (Figure 1). At the baseline, the heart rate of the EH_α_ was 80 bpm, his maximal heart rate was 294 bpm, and his recovery heart rate was 180.

### 3.2. NTproBNP at Rest, the Effect of Breed, Exercise and Training

The NTproBNP at rest (i.e., before exercise) was not significantly lower in the GSDs compared to the EHs at T_0_ (*p* = 0.074), but significantly lower at T_2mth_ (*p* = 0.045); it was also not significantly lower after exercise in the GSDs compared to the EHs at T_0_ (*p* = 0.063) and significantly lower at T_2mth_ (*p* = 0.027). At T_0_, exercise was associated with an increase in NTproBNP in both the GSDs (*p* = 0.016) and the EHs (*p* = 0.031); at T_2mth_, exercise again was associated with an increase in NTproBNP in the EHs (*p* = 0.031), but not in the GSDs (*p* = 1) (Figure 2). Higher maximal HR or higher ΔHR were not associated with higher NTproBNP concentrations after exercise or ΔNTproBNP. In one healthy EH (in addition to EH_α_), NTproBNP was above the reference range after exercise at T_0_ and before and after exercise at T_2mth_ (Figure 2 and Table 2).

The NTproBNP concentration of all dogs at rest correlated with multiple echocardiographic values (Figure 3). Specifically, it correlated positively with LVIDd, LVDDN, LA/Ao, and LVIDs. A negative correlation was observed with RWT and h/R. The correlations persisted after exercise for LVIDd, LVDDN, LA/Ao, LVIDs, RWT, and h/R. Finally, in the GSDs a strong positive correlation was observed between body weight and ΔNTproBNP at T_0_ (r = 0.87, *p* = 0.01).

A 2-month training period had no effect on NTproBNP at rest, neither in the GSDs (*p* = 0.578) nor in the EHs (*p* = 1). Within the groups, there was a significant change in NTproBNP after exercise at T_2mth_ compared to T_0_. However, whereas in the GSDs this value was significantly lower (*p* = 0.031), in the EHs it was significantly higher (*p* = 0.031) between the two time points, i.e., there was a training effect on post-exercise NTproBNP.

The ΔNTproBNP was not significantly different within groups (GSD, *p* = 0.297, and EH, *p* = 0.156) between the two examinations; between groups, however, whereas at T_0_ there was no difference in ΔNTproBNP (*p* = 0.180), it was significantly higher in the EHs at T_2mth_ (*p* = 0.018) (Table 2). 

### 3.3. Lactate Values at Rest, after Exercise and Effect of Training

At rest, the plasma lactate concentration was not different in the GSDs and the EHs, neither at T_0_ (*p* = 0.057) nor at T_2mth_ (*p* = 0.761). A 2-month training period did not have any effect on lactate concentration at rest, neither in the GSDs (*p* = 0.688) nor in the EHs (*p* = 0.688) (Table 3, Figure 4). Exercise caused a significant increase in lactate in the GSDs at T_0_ (*p* = 0.016) and at T_2mth_ (*p* = 0.031), while in the EHs, exercise caused a significant increase only at T_0_ (*p* = 0.031), but not at T_2mth_ (*p* = 0.156). Comparing lactate concentrations after exercise between the two examination times, there was no difference at T_2mth_ compared to T_0_ in both groups (EH *p* = 0.156; GSD *p* = 0.107). Between groups, Δlactate was not different, neither at T_0_ (*p* = 0.88) nor at T_2mth_ (*p* = 0.43) (Table 3).

The exercise-induced increase in the serum lactate concentration did not correlate with changes in ΔNTproBNP (T_0_ r = 0.144, *p* = 0.655; T_2mth_ r = −0.196, *p* = 0.541) and ΔHR (r = −0.0063, *p* = 0.984). No correlation was present between the the lactate and NTproBNP values after exercise (T_0_ r = 0.103, *p* = 0.749; T_2mth_ r = −0.339, *p* = 0.282).

## 4. Discussion

This study aimed to evaluate the effect of exercise and training on the serum NTproBNP concentration in dogs. Considering that this cardiac biomarker is connected to cardiac dimensions, these had to be assessed between the two dog breeds. The first important result is the fact that the EHs had significantly larger normalized left ventricles than the GSDs, in some EHs beyond the limit of normal (LVDDN < 1.6) [31]. However, heart size in the general EH group was not as large as in the specific overperforming dog EH_α_, and one GSD showed increased cardiac sizes.

The effect of breeding as opposed to training on cardiac size, i.e., a genetic as opposed to an acquired effect, has been shown in Greyhounds. Heart size was not different in racing Greyhounds compared to non-racing Greyhounds [32]. An effect of a 2-month short-interval and high-intensity training on heart size evaluated in the GSDs in this study was not evident.

The NTproBNP concentration at the baseline was higher in the EHs compared to the GSDs, but mostly still within the reference range (<900 pmol/L). A breed-associated difference in NTproBNP has been shown previously [11].

Studies comparing Greyhounds to other breeds have documented higher heart-to-body weight ratio [33], larger hearts on radiographs [34] and echocardiography [35], and higher NTproBNP serum levels [12]. If myocardial stretch or wall tension is the main stimulus for BNP release, one explanation could be inadequate wall thickness for the larger hearts in our EHs. The echocardiographic parameters of the heart size and wall thickness did correlate with NTproBNP; however, there was no difference between the EHs and the GSDs concerning LV h/r or relative wall thickness. Higher concentrations in the EHs may not be a pathological sign but simply a breed-associated difference. Likewise, Labrador Retrievers and Newfoundlands have previously been described as having higher concentrations than other breeds [11].

Exercise did have an effect on the serum NTproBNP concentration in both breeds performing different kinds of exercises. Whereas the increase was generally only mild and still in the range for dogs with no evidence of cardiac disease, in the individual EH concentrations were at a level considered gray-zone normal or even at a level consistent with cardiac disease.

Our study investigated the effect of training on the NTproBNP values in sports dogs. An investigation of healthy human beings evaluated the effect of training before running a marathon. People who had trained less had a greater increase in their NTproBNP values [36]. Another study evaluating cardiac biomarkers during training sessions and at the time of competition showed that athletes who had trained more demonstrated minor increases in NTproBNP [37]. In our study, nine weeks of training did not significantly affect the NTproBNP values at rest. However, a significant and opposite effect was present in the two breeds in the NTproBNP values after exercise. Unlike in the human studies, NTproBNP in the EHs was higher at T_2mth_, whereas in the GSDs it was lower. Besides a training effect in the GSD, the exercise intensity may have been lower at T_2mth_. Whereas maximal heart rates were comparable on the two occasions in the GSDs, the lactate concentration was lower after exercise at T_2mth_ in both groups (though not significantly); no clear explanation was found for this finding. To document an effective workout, ideally VO_2Max_ is measured, which was not possible in this study [38]. However, VO_2Max_ correlates well with lactate, and exercise is expected to cause an increase in serum lactate [39,40,41,42]. Heart rate is another parameter correlating well with VO_2Max_ in people [41]. In our experiments, exercise caused an increase in lactate and heart rates in most dogs at both time points, indicating that they underwent significant physical exertion. The serum NTproBNP values did not correlate with lactate or heart rate, potentially indicating that degree of individual exertion is not a central factor causing elevated wall tension.

There are several limitations to this study. First, the number of dogs was small, which may result in type 2 statistical errors, and due to the study design potentially to type 1 as well; this limitation may explain that, although a certain type of trend seems to be present (in NTproBNP, HR and lactate values), no significant difference was identified. Second, the comparison of the effects of different exercises, i.e., interval training versus endurance training, is flawed in this study because the dogs only performed either one. Ideally, the same breed would have accomplished both types of activities. Nevertheless, we could document that exercise does have a significant effect on NTproBNP in both breeds. Whereas this effect was considered clinically irrelevant in the GSDs, because concentrations were far below the level where cardiac disease might be suspected, it resulted in levels above normal in the individual EH. A further limitation of our study is that the dogs in the EH group did not have complete blood analyses, and therefore possible extracardial factors that may have influenced the NTproBNP behavior cannot be excluded. A central problem of our study is the lack of standardization of an intense exercise regime, because first, the training level of different dogs is expected to vary considerably, and second, it seems a very difficult task to assess how close individuals are approaching their limit of physical capacity during such a standardized exercise.

Finally, the very high NTproBNP concentrations in one particular EH (EH_α_) cannot be considered breed-associated normal. Either this was simply a statistical outlier or the wall tension was consistently elevated in this dog, even though objectively this dog was consistently overperforming over a long time span, making unlikely that he had a severe underlying cardiac disease as a cause of his increased values. A possible explanation for overperformance may be higher mental strength, i.e., going closer to his limits, suggested by relatively high lactate levels after exercise and higher HR_max_ and HR_rec_. Despite the overperforming achievements, a high NTproBNP value has to be considered as a possible marker of a subclinical heart disease and should trigger further investigations.

Differentiating athletic heart from true DCM in a dog with a DCM phenotype is a diagnostic challenge. However, it was not the goal of this study to answer this question. In future studies on this very question, the several parameters evaluated here merit closer examination: course of NTproBNP, heart rate, lactate levels before, immediately after, and 30 min after a standardized intense exercise, and additional biomarkers such as cTnI.

## 5. Conclusions

In conclusion, a two-month training period does not affect the NTproBNP concentration, but intensive physical activity does cause an increase in NTproBNP in dogs. In individual dogs, this increase may reach levels above the normal reference range.

## Figures and Tables

**Figure 1 animals-13-00016-f001:**
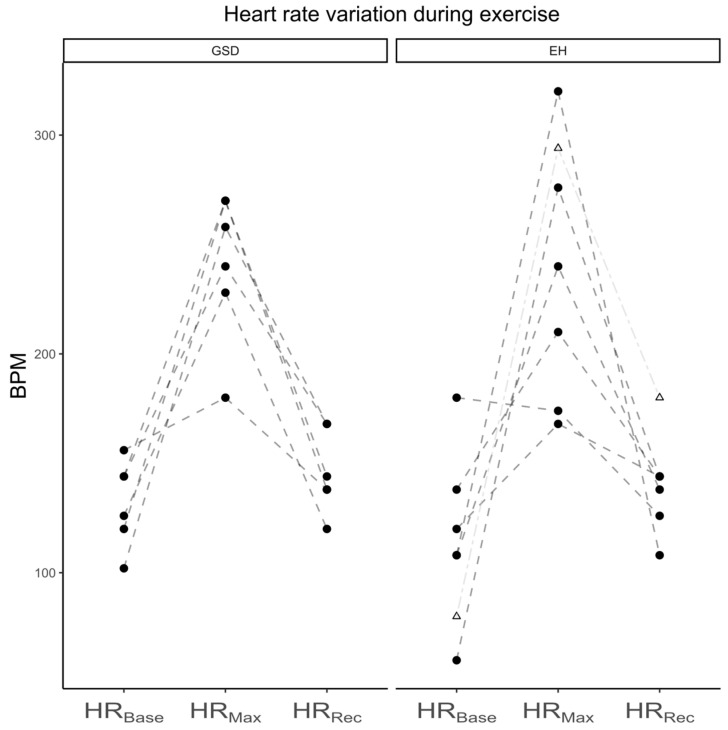
Heart rate variation during exercise in German shepherd dogs (GSDs) and Eurohounds (EHs) at T_2mth_. BPM: beats per minute; EHs: Eurohounds; GSDs: German shepherd dogs; HR_Base_: baseline heart rate; HR_Max_: maximal heart rate during activity; HR_Rec_: heart rate during recovery; Δ: values of EH_α_, outlier EH.

**Figure 2 animals-13-00016-f002:**
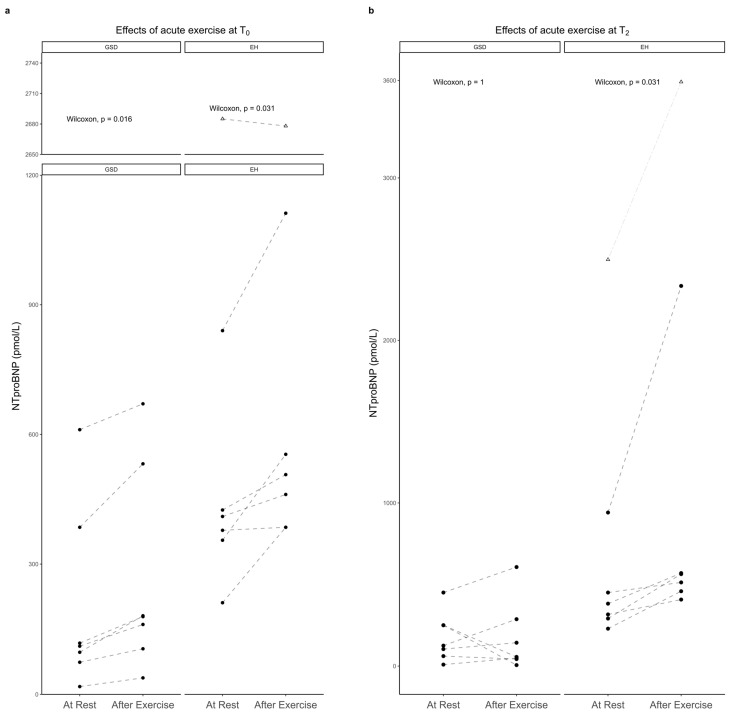
Effects of exercise on NTproBNP in GSDs and EHs at T_0_ (**a**) and T_2_ (**b**). EHs: Eurohounds; GSDs: German shepherd dogs; Δ: values of EH_α_, outlier EH.

**Figure 3 animals-13-00016-f003:**
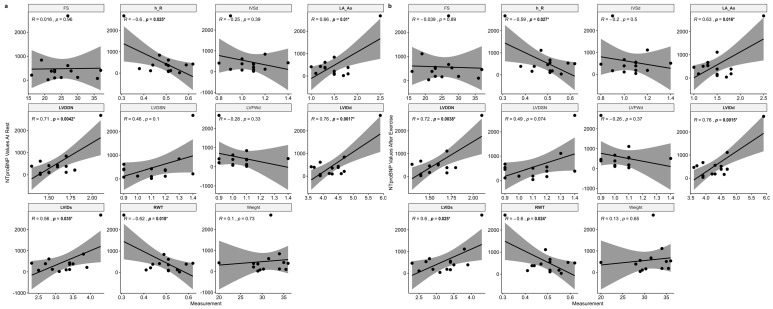
Correlation between echocardiographic parameters and NTproBNP values at rest (**a**) and after exercise (**b**). * Variables with significant associations.

**Figure 4 animals-13-00016-f004:**
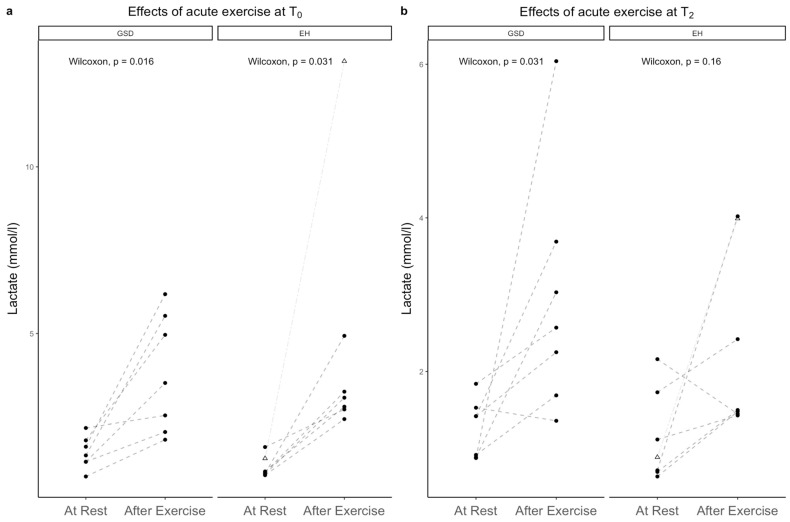
Effects of exercise on Lactate in GSD and EH at T_0_ (**a**) and T_2mth_ (**b**). EH: Eurohounds; GSD: German shepherd dogs; Δ: values of EH_α_, outlier EH.

**Table 1 animals-13-00016-t001:** Weight, age, and selective echocardiographic parameters in German shepherd dogs (GSDs) and Eurohounds (EHs) at T_0_.

	GSD (*n* = 7)	EH (*n* = 6)		
Variable	Median (Range)	Median (Range)	*p*-Values	EH_α_ *
Bodyweight (kg)	31.5 (29–36)	28.5 (20–35)	0.22	32
Age (years)	3 (2–4)	6 (1- 10)	0.26	7
LA/Ao	1.5 (1.1–1.8)	1.4 (1.0–1.8)	0.63	2.5
IVSd (cm)	1.1 (0.9–1.2)	1.1 (0.8–1.4)	0.65	0.9
LVIDd (cm)	3.9 (3.7–4.6)	4.5 (3.6–4.7)	0.16	5.9
LVWd (cm)	1.1 (1.0–1.1)	1.05 (0.9–1.4)	0.76	0.9
LVIDs (cm)	2.9 (2.5–3.4)	3.45 (2.3–3.9)	0.15	4.3
FS%	27 (21–36)	22.5 (16–37)	0.3	27
LVDDN	1.4 (1.3–1.7)	1.65 (1.5–1.8)	0.01	2.1
LVDSN	1.0 (0.9–1.2)	1.2 (0.9–1.4)	0.08	1.4
h/r	0.51 (0.43–0.59)	0.485 (0.38–0.62)	0.42	0.31
RWT	0.51 (0.41–0.59)	0.48 (0.43–0.62)	0.47	0.31

LA/Ao, left atrium to aortic root ratio; IVSD, interventricular septum diameter in diastole; LVIDd/LVIDs; left ventricular internal diameter in diastole/systole; LVWd, left ventricular free wall diameter in diastole; FS%, fractional shortening; LVDDN/LVDSN, LVIDd/LVIDs normalized to body weight; h/r, wall thickness/radius, i.e., LVWd/½LVDd; Relative Wall Thickness (RWT), i.e., (LVWd + IVSd)/LVDd; * data of one statistical outlier EH.

**Table 2 animals-13-00016-t002:** Effect of intensive exercise on NTproBNP serum concentration (pmol/L) in GSDs and EHs, before (T_0_) and after a training period (T_2mth_).

Group	Rest Median (Range)	Exercise Median (Range)	ΔNTproBNP Median (Range)	*p*-Value
GSD T_0_	111 (17–611)	179 (38–671)	60 (20–147)	0.016
GSD T_2mth_	125 (9–450)	56 (6–606)	39 (−244–162)	1
EH T_0_	394 (211–840)	484 (385–1112)	128 (7–272)	0.031
EH T_2mth_	349 (291–941)	538 (407–2335)	208 (62–1394)	0.031
EH_α_ * T_0_	2685	2678	−7	
EH_α_ * T_2mth_	2497	3591	1094	

GSD, German Shepherd Dog; EH, Eurohound; ΔNTproBNP, NTproBNP values after minus before exercise; * data of one statistical outlier EH.

**Table 3 animals-13-00016-t003:** Plasma lactate concentration (mmol/L) in GSDs and EHs before and after exercise, before and after a 2-month training period.

Group	Baseline Median (Range)	Exercise Median (Range)	ΔLactate Median (Range)	*p*-Values
GSD T_0_	1.4 (0.7–2.2)	3.5 (1.8–6.2)	2.4 (0.4–4.6)	0.01563
GSD T_2mth_	1.4 (0.9–1.8)	2.6 (1.4–6.0)	0.8 (−0.2–5.2)	0.031
EH T_0_	0.8 (0.8–1.6)	2.2 (2.1–4.3)	2.1 (1.1–4.1)	0.031
EH T_2mth_	0.92 (0.6–2.2)	1.49 (1.4–4.0)	0.745 (−0.7–3.3)	0.156
EH_α_ * T_0_	1.3	13.2	11.9	
EH_α_ * T_2mth_	0.9	4.0	3.1	

GSD, German Shepherd Dog; EH, Eurohound. ΔLactate, after minus before exercise; * data of one statistical outlier EH.

## Data Availability

The datasets used and/or analyzed during the current study are available from the corresponding author upon reasonable request.

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
