# Peer review of "Effects of Breed, Exercise, and a Two-Month Training Period on NT-proBNP-Levels in Athletic Dogs"

_animals, 2022, doi:10.3390/ani13010016_

Round 1

Reviewer 1 Report

PDF file with comments attached here.

Author Response

We would like to thank the reviewer for the excellent amount of interesting feedback provided. We answered the comments in a separated file.

Reviewer 2 Report

Reviewers comments

Effects of breed, exercise, and a two-month training period on 2 NT-proBNP-levels in athletic dogs

In general:

This is an interesting paper. It is quite difficult to compare these two populations with another because of their different age and training conditions before the study period. I think some aspects should be discussed more into detail, and there are some small aspects tob e corrected. In general it is well writtened paper and the literature review at the beginning is really nice. After performing these corrections it should be published.

Abstract

Introduction:

The introduction is a really nice review of the literature and describes the reasons fort he performed study quite well.

Material and methods:

Line 116: In this specific dog: Beside NT-pro BNP and echo does other tests were performed like Troponin I or an Holter EKG? If there is a volume overload oft he left ventricle other possible diseases like intrapulmonary shunt connections are ruled out?

Line 140: Is there a specific reason why echo and Holter were performed only at specific time points?

Line 159: When the possebility of a volume increase in dogs with an athletic heart is described, like in the one dog is there a performance and calculation of systolic functional parameters not also needed?

Line 177: To really interpretate the NT-pro BNP levels obtained in the study the possible secondary effects of other organs or diseases must be ruled out, like described in the nice introduction. So it would be necessary to have a complete bichemical profile with renal values and also a blood pressure measurement at baseline is also necessary 

REsults:

Line 203:

What was the heart rate of the both populations at the echo examination? Was there a difference of the normalised left ventricle when calculted to the specific personal heart rate, because it is described that there are differences and in the population of EH dogs it could be the fact that they are lower in heart rate because they are trained before.

Discussion:

Line 311: I think it is not really fair, because I would expect that in the population of trained dogs is an effect and here we only compare it to one breed like Greyhound, and we know that in this specific breed a lot oft hings are different. Fromm y point of view the consequence would be to compare a group of non-trained EH dogs with a trained population. The other fact ist hat there is difference in the age of both populations. It would be great to discuss these facts additionally.

Line 328: I think it is also not fair to call the type of exercise in the german shepard group „intensive“ in relation to that in the other group.

Line 351: The different values of NT-pro BNP in both breeds after the training period should also be discussed with the type and intensity of exrecise both populations did. (It is done in the limitations part)

Line 368: I think not only overperforming should be discussed here, also a subclinical heart disease could be the problem here.

Line 380: Yes it produces an increase, but I think it should mentioned that it is an clinical not relevant increase.

Author Response

Dear Reviewer,

We look forward to hearing from you in due time regarding our submission and to respond to any further questions and comments you may have. You can find our answers to your points mentioned in the attached pdf.

Reviewer 3 Report

This study presents the changes in NT pro BNP in different breeds of healthy dogs under different exercise intensity conditions. The manuscript is well written, the protocol performed is correct according to this reviewer and the conclusions support the results. I believe that this study bring novel information to the natriuretic hormone “behavior” during different conditions in healthy dogs.

I have some minor questions regarding the result presentation, which I think that need to be clarified:

Line 224: Authors state that maximal HR in the EH (225; 168 – 320) were not significantly higher than baseline (114; 60-180), reporting a P-value of "0.063". It is interesting how this is not different since the maximal median HR is almost double as compared to baseline. If this is not a typing error, please add a clarification discussion for the reader to better understand whether this result was potentially induced by the small sample size or anything else. 

Line 237 – authors state that The NTproBNP at rest (i.e. before exercise) was lower in the GSD compared to the EH at T0 (p = 0.074), however the P value does not reach the significance level and the same thing after exercise. Did authors mean that the value is just lower or there was a statistically significant value and “0.074” was a typing error? Could authors clarify this statement?

Line 322 and 362 – Instead of EU, did authors mean EH?

Author Response

Once again, we would like to thank the reviewer for taking the time to examine our manuscript. We have uploaded a separate file in order to be able to adequately answer the individual points requested.
